# Position: Continual Learning Should Move Beyond Incremental Classification

## Abstract

Continual learning (CL) is the sub-field of machine learning concerned with accumulating knowledge in dynamic environments. So far, CL research has mainly focused on incremental classification tasks, where models learn to classify new categories while retaining knowledge of previously learned ones. Here, we argue that maintaining such a focus limits both theoretical development and practical applicability of CL methods. Through a detailed analysis of concrete examples — including multi-target classification, robotics with constrained output spaces, learning in continuous task domains, and higher-level concept memorization — we demonstrate how current CL approaches often fail when applied beyond standard classification. We identify three fundamental challenges: (C1) the nature of continuity in learning problems, (C2) the choice of appropriate spaces and metrics for measuring similarity, and (C3) the role of learning objectives beyond classification. For each challenge, we provide specific recommendations to help move the field forward, including formalizing temporal dynamics through drift processes, developing principled approaches for continuous task spaces, and incorporating density estimation and generative objectives. In so doing, this position paper aims to broaden the scope of CL research while strengthening its theoretical foundations, making it more applicable to real-world problems.

## 1. Introduction

While textbook machine learning methods assume data distributions are stationary and all training data is collected upfront, in many practical applications, new data becomes available, and new requirements (tasks) emerge over time.

[1]Anonymous Institution, Anonymous City, Anonymous Region, Anonymous Country. Correspondence to: Anonymous Author <anon.email@domain.com>.

Preliminary work. Under review by the International Conference on Machine Learning (ICML). Do not distribute.

Learning then becomes a continual process, updating model parameters all the time to keep track of the changing conditions. The non-stationarity of this 'incremental classification' setting —be it due to the new tasks resulting in new loss terms or due to shifts in the data distribution (aka domain shifts)— makes standard methods fail, resulting in 'catastrophic forgetting' of previously learned knowledge. In contrast, the goal of continual learning methods is to accumulate knowledge without such catastrophic forgetting.

Continual learning (CL) is a broad framework primarily explored in research papers through the lens of classification. The dominant setup consists of a sequence of classification tasks, usually obtained by taking a classification benchmark dataset and splitting it into smaller parts, referred to as 'tasks', each containing data exclusively from a disjoint subset of classes. When learning a task, it is assumed that only the data of the current task is accessible. This setup is chosen for its high reproducibility, transparency, and simplicity. Many CL methods are evaluated and compared only in this setup, encouraging overfitting to or even designing specifically for this particular setup. It is implicitly assumed that conclusions derived from this setup and algorithms designed for it generalize to more practical use cases and other tasks beyond classification. But is that really the case?

**In this position paper, we argue that moving beyond the incremental classification paradigm is crucial for developing CL methods that are theoretically grounded and broadly applicable to real-world problems.** While we indeed acknowledge the utility of addressing incremental classification, we argue that such solutions may not generalize as well as often implicitly assumed. In particular, many works claim "state of the art" results in CL while only considering incremental classification. To this end, we highlight the limits of methodology developed solely in the context of supervised classification by examining concrete examples involving multi-target classification, optimization with constrained output spaces, CL in the absence of a natural discretization of tasks, and higher-level concept memorization. We combine these with conceptual analysis of prototypical continual learning methods like iCaRL (Rebuffi et al., 2017) and regularization-based ones like EWC (Kirkpatrick et al., 2017) or moment matching (Lee et al., 2017). By illustrating challenging scenarios where CL is particularly relevant, we highlight potential pitfalls when applying naïve

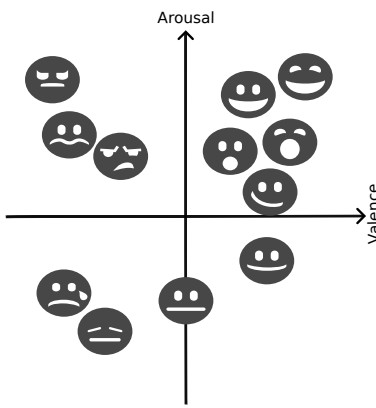

*Figure 1.* Map of diverse facial expressions on Arousal-Valence axes. This representation captures the inherently continuous variation of expressions as opposed to, e.g., "angry" and "sad".

implementations to our selected examples. This approach provides valuable insights for the CL research community, guiding future research directions.

We proceed as follows. We start by examining concrete examples that illustrate key limitations of current CL approaches. We then analyze fundamental conceptual challenges these examples reveal. Finally, we consider alternative perspectives and conclude with recommendations for future research.

## 2. Core Examples

In the following subsections, we consider important example problems, each illustrating an extension of classic supervised CL. In each case, to illustrate the importance of considering extension, we consider the difficulties in applying the popular pillars of CL methodology: functional approaches, regularization strategies, and data retention, respectively (*i. e.*, iCaRL and Knowledge Distillation, Elastic Weight Consolidation, Coresets). We close each subsection with suggestions for future directions of CL research to address these difficulties.

### 2.1. How well addressed is supervised CL for classification?

To examine generalizability in familiar territory, we start with an example close to standard class-incremental supervised learning. Specifically, we consider the problem of continual facial expression detection and classification from image data using neural networks.

A common representation for facial expressions uses 12 Action Units — discrete facial muscle regions that can be active or inactive. While more detailed representations indeed exist, we consider this simple case only. The core

challenge here is that it is actually a multi-target prediction problem. While it is a classification problem, the archetypical CL problem has a single set of discrete clusters, which we identify as classes. It is not clear how to adapt CL methods that rely heavily on such clustering to this problem. Further, requiring the presence of explicit classes requires an explicit discretization of the problem into clusters. We argue that for data such as facial expressions, such a discretization is at best difficult to correctly construct, and at worst incoherent. If we were to formulate our problem in terms of regression, such as in the 2D arousal-valence representation of expressions seen in Fig. 1, such discretization issues would disappear. This, however, would explicitly restrict us to CL methods that function correctly in the absence of class labels.

Even if we were to naïvely apply a popular method that uses class labels to this problem, *e. g.*, iCarl (Rebuffi et al., 2017), we would encounter similar issues. Specifically, iCarl populates a memory buffer such that it is balanced across classes and the mean of the examples $p_j$ for any given class is close to the mean $\mu$ for the cluster $X$ of datapoints $x$ corresponding to that class:

$$p_k \leftarrow \arg\min_{x \in X} \left\| \mu - \frac{1}{k} \left[ \phi(x) + \sum_{j=1}^{k-1} \phi(p_j) \right] \right\|, \quad (1)$$

where $\phi$ is some reasonable feature representation of datapoints. If one treats the entire dataset as a single cluster, then we do not expect the center of that cluster to be meaningful, as the distribution is highly multimodal. Alternatively, one could use the multi-target classes and consider a single cluster for the purposes of iCarl to correspond to a choice of class for every target (every Action Unit). Unfortunately, even in this case, with only two classes per target, the total number of clusters grows exponentially in the number $N$ of targets (Action Units) as $2^N$. While this may still be possible for the case of 12 Action Units and, in turn, 4096 total clusters, it will quickly explode combinatorially as $N$ increases; 32 Action Units would give already $4.3 \times 10^9$ clusters, likely exceeding the total dataset size by orders of magnitude and making the idea of taking the average of the datapoints within a cluster impossible.

The traditional classification-based settings of CL encourage methods to explicitly rely on class labels, and this implicitly requires the data to be discretized into a single sensibly-sized clustering. We have seen that alternatives such as multi-target classification bring their own problems, and that more continuous regression formulations of the prediction task may remove these difficulties. We further note that the cross-entropy loss itself introduces difficulties for class-incremental learning, due to the necessity to add new output nodes, and, more generally, due to the non-constant curvature of the cross-entropy loss interacting poorly with the implicit gaussian posteriors of EWC-like parameter regular-

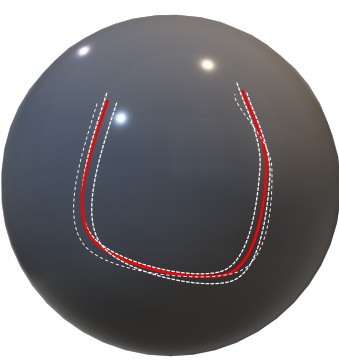

*Figure 2.* Depiction of a trajectory learned from demonstrations (shown in red) on the surface of a sphere. This example illustrates the challenge of constrained structured prediction in robotics, where valid outputs must lie on a two-dimensional manifold (the sphere's surface) within a three-dimensional space. Traditional continual learning approaches using Euclidean distance metrics may fail to maintain such geometric constraints during learning.

ization methods. We expect that there are many applications for CL where the assumption of a single-target classification objective artificially complicates CL, and results in CL methods not generalizing as well as they should.

### 2.2. How can CL accommodate constraints?

Robotics presents another domain where naive applications of common CL methods can be unreliable. In robotics, problems often involve predictions lying in nonlinear output spaces due to physical constraints imposed by a robot's embodiment and environment. Directly minimizing a loss measured in a Euclidean output space may fail to capture the structure of the output, compromising the optimality of control outputs and potentially violating safety constraints. While substantial progress has been made on structured prediction for robotics in non-continual settings, extending these approaches to continual learning remains largely unexplored.

Consider the task of generating robot arm trajectories constrained to lie on the surface of a sphere, for example, to ensure safety by avoiding collisions (Fig. 2). The full space of end effector locations is three dimensional, but the space of valid outputs is the two dimensional spherical surface. Methods have been proposed to constrain model outputs to such structured target spaces, such as encoding the output space into a linear surrogate space for training, and then decoding predictions back into the original structured space (Bakır et al., 2007). This can also be done implicitly with surrogate losses that enforce desired output properties (Ciliberto et al., 2020), an approach used in imitation learning

(Zeestraten et al., 2017; Duan et al., 2024) and reinforcement learning (Liu et al., 2022). However, the feasibility of extending this approach to continual robot learning remains understudied. A pioneering work in this direction is (Daab et al., 2024), introducing a method for incrementally learning motion primitives on Riemannian manifolds.

If one were to naively apply a parameter-space regularization method, such as EWC (Kirkpatrick et al., 2017), to a task with manifold constraints on the outputs, the approach would minimize the squared distance in parameter space between old and new parameters, weighted by their importance. Specifically, one is assuming that the increase in loss for task $t$ as the parameters $\theta$ drift from their optimum $\theta_t^*$ in future learning is approximately proportional to

$$\mathcal{L}_t(\theta) - \mathcal{L}_t(\theta_t^*) \propto \sum_{i=1}^{|\theta|} F_{tii}(\theta_i - \theta_{ti}^*)^2, \qquad (2)$$

where $F_{tii}$ is the diagonal of the Fisher matrix measuring the relevance of particular parameters $i$ to task $t$. Unfortunately, it is likely that, even if the predictions at $\theta_t^*$ obey the manifold constraints, the predictions of some arbitrary $\theta$ which is merely close to $\theta_t^*$ according to the Fisher matrix will not. If the manifold constraints represent, for example, a safety constraint, this is clearly unacceptable behavior for a CL algorithm. Not only should each task optimum satisfy the constraints for that task, but the CL algorithm must maintain their satisfaction throughout further training next to minimization the loss.

In summary, CL methods tend to focus on ensuring that future outputs remain "close" to past outputs, and assume that sufficiently close outputs will remain valid. In the presence of manifold constraints, it is clear that a naive distance measure on the full output space will not be sufficient to hold future outputs within the valid range. We expect that exploitation of the surrogate loss approach may allow parameter regularization, functional regularization and simple memory buffer-based CL methods to potentially generalize to the structured prediction setting. But this generalization must be demonstrated, and, in cases where this surrogate loss is not provided, the relevant problems compensated for in some other way.

### 2.3. What is a task? CL in continuous domains.

In the classic case of CL we have either a single task with a growing number of classes or a discrete set of tasks; here, the term "task" typically refers to a context in which an input-output pair can be assigned a loss. For example, one might consider classifying whether an MNIST digit is prime as one task, and classifying whether it is divisible by 3 as another. Alternatively, one could progressively introduce new classes within the same task, *i.e.*, the classic class-incremental setting. These standard CL settings are tautologically discrete, but are discrete changes the only ones we

should be concerned with in CL?

For the sake of illustration, consider the problem of pushing a box along the ground using a single manipulator, *i.e.*, force can be applied at a single location on the box. Now, allow the box to contain different arrangements of items such that its internal weight distribution varies. Imagine solving this problem as a human. You will instantly understand that you would need to apply force along a line passing through the box's center of mass; otherwise, the box would rotate instead of sliding forward. Further, if the content of the box is not visible, a human can infer the location of the center of mass from the way the box reacts to being pushed and adjust their strategy to compensate. Clearly, the correct action varies depending on the weight distribution of the box, but how can this be formulated within the incremental classification framework? Not only is the space of "tasks", *i.e.*, the center of mass locations continuous, but there is no task label, and the task should be inferred from context. One could argue that, since this inference from context is possible, there is only one task with instead multiple classes, but then the problem again recurs when trying to discretize into classes. The robot may, however, encounter novel regions of weight distribution space, and it is desirable to transfer knowledge about such regions across time, indicating the presence of some task-like or class-like continual element. The aggregation of policies across this continuous "task" space is thus a natural thing to attempt, and is a problem to which CL should ideally offer solutions.

One can naively imagine applying knowledge distillation (Hinton et al., 2015) to this box pushing problem. For instance, the loss $\mathcal{L}$ due to Li & Hoiem (2016) generalizes in the presence of a memory buffer to the following:

$$\mathcal{L}(\theta) = \sum_n D_{\text{KL}}(T_n||P_n(\theta)) + \lambda \sum_b D_{\text{KL}}(T_b||P_b(\theta)) \tag{3}$$

consisting of two KL-divergence terms between target distributions $T$ and predicted distributions $P(\theta)$. For new data, the targets $T_n$ are perfectly confident ground truth probabilities, whereas targets $T_b$ for data in the memory buffer are set to the original predicted distribution when this data was memorized. $\lambda$ is a hyperparameter which allows prioritization between new and old data, and we have omitted a temperature parameter (*i.e.*, implicitly set it to one) from the buffer term for simplicity. Suppose that over time the distribution of weight distributions encountered by our robot shifts. The immediate difficulty is that it is non-trivial to distinguish new tasks from old tasks, as the true boundaries are fuzzy. If we regularize the learned function weakly, the model will forget weight configurations encountered only in the old data, but if we regularize strongly then it will be unable to improve its performance on weight configurations more common in the new data. Rather than simply holding the function stable on old data and allowing it to drift on

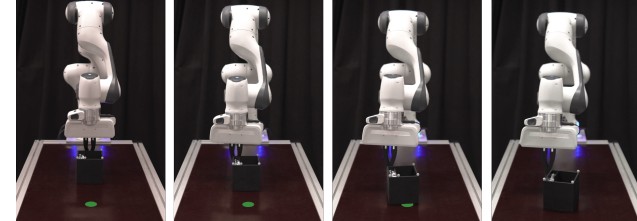

*Figure 3.* A robot arm pushing a box onto a target marker (green). The arm makes contact at a single point and must adjust for the weight distribution in the box. Image from (Tiboni et al., 2024).

novel points, it is necessary that the function be regularized to different degrees in different regions even if they have been encountered before. In particular, it is no longer true that lower drift on all old datapoints is always better — some amount of "forgetting" is desirable in order to improve behavior in scenarios where the model fit is imperfect due to sparse but extant data.

The problem of continual learning in a real world setting where novel classes or corrupted sensor data may be present and must be handled correctly is more broadly referred to as Open World learning (Mundt et al., 2023). An existing angle of attack on the problem of unmarked task or class boundaries is Out-of-Distribution (OOD) detection (Hendricks & Gimpel, 2017; Liang et al., 2018; Sastry & Oore, 2020; Sun et al., 2021; Liu et al., 2020; Huang et al., 2021; Francesco Cappio Borlino, 2022), which focuses on identifying samples which deviate from the previously seen distribution due to the presence of a discrete distribution shift. Unfortunately, our problem here is deeper — the discrete clusters or distribution shifts which OOD detects are not merely unlabelled, but nonexistent. Looking forward, we argue that the notion of "task" in the classic incremental setting must be generalized, not only to cases where the task labels are implicit rather than explicit, but to cases where no discrete task label can coherently be assigned due to the continuous nature of the task space.

### 2.4. What is memorable?

The classic continual learning paradigm focuses on retaining input-output pairs, a natural approach for avoiding catastrophic forgetting. However, humans also retain more abstract forms of knowledge, suggesting that this input-output paradigm may be insufficient (Ilievski et al., 2024). We explore this issue in the context of reinforcement learning (RL), where the expense of gathering data makes memory especially valuable.

RL memory buffers typically store concrete state-action-result tuples. However, humans also remember more abstract information, such as the availability of strategies. Con-

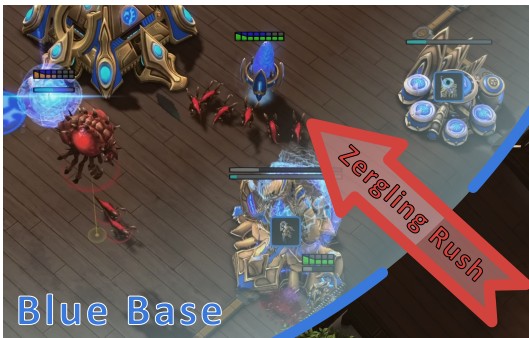

*Figure 4.* Zergling rush in *Starcraft II*: the blue player (with the tan/blue buildings) has failed to completely block the entrance at the lower right, allowing zerglings (small and red) into their base.

sider the so-called "zergling rush" in Starcraft II (Fig. 4): if zerglings infiltrate the opponent's base early on, they can quickly win by destroying the opponent's economy. To prevent this, players position their buildings to act as walls in order to block their entrances. The mere possibility of a zergling rush, even if rarely executed, deeply shapes the game. Humans remember this strategic principle —how can we capture this sort of abstract knowledge in CL systems?

Coreset methods (Bachem et al., 2015) illustrate the limitations of focusing solely on concrete examples. A coreset is a weighted subset of the whole dataset which achieves some particular metric of performance, and is usually optimized to be as small as possible. For example, Mirzasoleiman et al. (2020) consider the smallest set $S$ which, given weights $\gamma_j$, results in total loss gradients $\nabla\mathcal{L}(\theta)$ within $\epsilon$ of the total gradient for the whole dataset $D$ for all parameter values $\theta$ of a given model:

$$S^* = \arg\min_{S\subseteq D,\gamma_j\geq 0}|S|, \text{ s.t.}$$
$$\max_{\theta\in\Theta}||\sum_{i\in D}\nabla\mathcal{L}_i(\theta) - \sum_{j\in S}\gamma_j\nabla\mathcal{L}_j(\theta)|| \leq \epsilon \quad (4)$$

The problem here is that the underlying method, say, a model-free reinforcement learning algorithm such as Soft Actor-Critic (Haarnoja et al., 2018), does not natively know how to reason about strategic counterfactuals. Concrete examples of the zergling rush being used against an opponent who has not walled off will be very sparse during optimal self-play, so the contribution to total gradients in such data may be low. Further, if the underlying RL algorithm requires many examples to reliably learn the universal availability of the strategy, individual examples would likely not improve the gradient approximation for other examples very much. Thus, even if there is a noticeable total gradient contribution corresponding to rare actual zergling rushes, the size of a coreset which included the relevant examples might be impractically large. This becomes even clearer when we look at techniques inspired by explainability methods (Gilpin

et al., 2018; Burkart & Huber, 2021), such as Prototype Networks (Chen et al., 2019). Adapted for CL (Rymarczyk et al., 2023), the heuristic for buffer population would be "store those examples most relevant to decisions." Clearly, if the underlying method is unable to sufficiently generalize to correct decisions from individual concrete examples, or there is no actual concrete example available, then this whole class of methods cannot solve our problem. The problem here is the fundamental difficulty of compressing high level concepts like this availability of a strategy (*i. e.*, systematic counterfactual use as opposed to occasional actual use) into a memory containing only real concrete examples.

The high level problem of remembering something more abstract than raw data, is, of course, not a new one. Indeed the Never Ending Learners of Chen et al. (2013); Mitchell et al. (2018) integrate varied information sources into a database of abstract relational beliefs. Further, humans constitute an existence proof of the feasibility of such a heterogeneous memory architecture in biological neural networks (Marr, 1971; McClelland et al., 1995). Even when constrained to considering only long-term memory in particular, multiple components can be distinguished, such as episodic, semantic, and procedural memory (Tulving & Donaldson, 1972; Graf & Schacter, 1985). Nor is it the case that richer notions of memory are unknown to contemporary work on artificial neural networks (Thorne et al., 2020). We argue that this problem of remembering higher level information should be revisited in the contemporary CL context.

## 3. Conceptual Framing: Where from here?

Having examined several illustrative examples that highlight the limitations of naive applications of current continual learning approaches, we now turn to a systematic analysis of three key conceptual challenges that must be addressed to move the field forward. We structure our discussion around three fundamental aspects: the nature of continuity in learning problems, the choice of appropriate spaces and metrics for measuring similarity, and the role of local objectives in learning. For each aspect, we first present key considerations that emerge from our analysis, followed by specific recommendations for future research directions.

### 3.1. On Continuity (Cont.)

**Considerations: Cont.** When designing CL systems, one must examine what continuity means and how it manifests. Two fundamental forms of continuity shape the space of possible approaches: temporal continuity in how tasks evolve, and continuity in the underlying task space itself. These distinct types of continuity create different constraints on learning algorithms and require different treatment.

**Cons #1: Temporal continuity.** We will refer to a change over time of the joint distribution of data points and prediction targets as "drift". This is classically handled by assigning a potentially different distribution $\mathcal{D}_i$ to every data point $x_i$ (Gama et al., 2014), with drift occuring when $\mathcal{D}_i \neq \mathcal{D}_j$. We advocate here for the approach of Hinder et al. (2020), who propose a drift process by associating each datapoint $x_i$ with a time $t_i$, such that two datapoints sharing a time also share the same distribution. The distributions $D_t$ are defined as Markov kernels in the time domain, and it is now possible to postulate limiting statements similar to the batch setup or discuss concepts such as the mean distribution over a period of time.

**Cons #2: Task continuity.** While the comparatively simple case of continuously varying mixing coefficients of a discrete task set has been considered under the name "task-free continual learning" (Lee et al., 2020; Jin et al., 2021; Shanahan et al., 2021), the possibility of a truly continuous task set has been raised (van de Ven et al., 2022), but we are not aware of a systematic exploration of this setting. For example, in the task-free setting one might first infer task identity and then use task-specific components (Heald et al., 2021), but even if task identity inference is solved, the lack of a discrete task set in the harder case makes the use of task-specific components no longer trivial.

> **Recommendations: Cont.** Based on our analysis of continuity challenges in CL, we propose three key directions for future research: 1) formalizing temporal dynamics through drift processes rather than pointwise distributions, 2) understanding and managing the impact of data presentation schedules, and 3) developing principled approaches for handling continuous rather than discrete task spaces. These recommendations aim to help the field move beyond implicit assumptions about continuity toward more theoretically grounded methods.

**Rec #1: Use drift processes.** We recommend working with Drift Processes over datapoint-indexed distributions, as this makes the temporal structure explicit. In particular, the extent to which temporally close distributions are expected to be similar must be assumed explicitly rather than implicitly. We believe the associated improvement in clarity of thinking will enhance future work.

**Rec #2: Consider schedule dependence.** Formalize a data stream as an underlying dataset and an order in which this is presented, or *schedule*. While known in stream learning (Gama et al., 2014), the effects of such a schedule are considered explicitly only by relatively few CL works (Yoon et al., 2020; Wang et al., 2022), and Wang et al. (2022) showed that most existing continual learning algorithms suffer drastic fluctuations in performance under different schedules.

After considering the expected temporal correlations of the data stream via drift processes, it is likely that significant permutation symmetries (*e. g.*, discrete task orderings) will remain. After establishing which permutations of the stream do not constitute meaningful information from which the model should learn, future work should strive to maximize invariance of CL algorithms to such permutations.

**Rec #3: Towards continuous task identity.** Finally we note that, while discreteness of the underlying task set has been an important and productive underlying assumption in continual learning research, principled methods of handling task identity in the truly continuous case (*e. g.*, section 2.3 and maybe even 2.4) should be developed. Task-specific components, for example, should still be possible where task identity is not discrete. When representing a task as, *e. g.*, some embedding in a continuous latent space, however, they are no longer trivial and are indeed interestingly non-trivial. Such principled approaches should strive to account for the now much richer geometry of the task space.

### 3.2. On Spaces

> **Considerations: Spaces.** When examining CL systems, we encounter three distinct types of continuous spaces: parameter space, data space, and function space. Each of these spaces requires careful consideration of how to measure "similarity" or "distance" - a choice that is sometimes forced by the problem structure. Even after selecting a space, the choice of metric remains critical, as different metrics can capture different aspects of the learning problem. Some scenarios may even require inherently asymmetric measures of similarity.

**Cons #1: The three common spaces.** Most obviously, we have the continuous space of parameters. Often we also have a continuous space of possible data items, *e. g.*, arrays of floating point pixel values. Finally, we have the continuous space of functions representable by our neural network. If we identify a "task" with "the mapping from inputs to outputs which solves the task" then it can be seen as a special case of a function space.

When one needs to measure "similarity" or "distance" in continual learning, one will in general do so in one of these spaces. Sometimes this is a choice, sometimes it is forced. For example, when considering a mixture of experts solution to a variety of tasks where the architectures of the neural network models corresponding to the experts differ, it is impossible to measure distance in parameter space. In this case we must instead consider function space.

**Cons #2: Metrics.** Even once the choice of space is made, "distances" are not determined until we choose a metric on

that space. Sometimes there will be a natural choice (*e. g.*, the Fisher metric in function space for classification tasks, or more generally for tasks where the output is a probability distribution). In an application such as weight space regularization, there is a simple choice of the Euclidean metric, but this choice is inherently incapable of identifying more or less important parameters for a given task, and may even violate safety constraints in a case like that of section 2.2. The more expressive choice of the Fisher metric as used in Natural Gradient Descent would allow such parameters to be identified. This may allow a new task to make use of those subspaces of parameter space left unspecified by the preceding tasks.

**Cons #3: Divergences.** Finally, it is often the case that a notion of "distance" in a continual learning problem can be identified with a KL divergence, and is thus inherently asymmetrical. For example, suppose we wish to identify new tasks by measuring the "distance" between a memory buffer and a sequence of new datapoints. If the memory buffer contains datapoints from tasks A and B, but the sequence of new datapoints comes only from task B, is it a new task? Clearly not. But if this was reversed, and the new datapoints came from A and B, while the buffer came only from B, then the memory buffer would be insufficient to determine correct behaviour on the new points from task A and the answer to the question "is there a new task" must be yes. Consider the case of two 2D Gaussian distributions centered at $(0,0)$ and $(1, 0)$ with isotropic standard deviations 2 and 0.5, respectively. The KL divergence in one direction is 2.8 bits, but in the other it is 20.5 bits. Intuitively, this is because samples from the small Gaussian are in-distribution for the large Gaussian, but not vice-versa. More concretely, in the previously considered application of the Fisher metric to parameter space regularization, one direction corresponds to measuring distances relative to the Fisher metric measured on the new datapoints, whereas as the other corresponds to using the Fisher metric measured on the buffer.

> **Recommendations: Spaces.** Based on our analysis of the different spaces and metrics in continual learning, we propose several practical guidelines for developing more effective methods. These recommendations focus on making explicit choices about spaces and metrics, recognizing potential asymmetries in similarity measures, and considering alternative spaces when standard approaches fail.

**Rec #1: Choose the correct metric and space.** Firstly, one must choose the space in which to measure this similarity or distance. The straightforward option might be to consider raw data such as pixel values, but perhaps semantic differences would be easier to detect in some function space, such as a latent space of a neural network. Then, having

identified the correct space, one must choose a metric on that space. Even when making "no choice" and using the Euclidean metric, one should be mindful of what this means. For example, when doing weight space regularization, using a quadratic penalty in the Euclidean metric corresponds to the assumption that the appropriate posterior on weights is an isotropic Gaussian. Making the implications of this "non-choice" concrete will allow the implicit assumptions to be sanity-checked.

**Rec #2: Remember that the correct notion of similarity may not be symmetric.** One should also pay attention to any asymmetries in the application of a notion of distance. Often the "distance" measure required in an algorithm will correspond to a KL divergence. Whether you would like your distance measure to behave like forward KL divergence or reverse KL divergence depends on the purpose of the measure: "how informative is task A about task B" will often have a different answer to "how informative is task B about task A". Choosing the wrong direction here will likely result in severe algorithm underperformance, even though both directions agree when the tasks being compared are relatively similar. Since asymmetries here become most salient when similarity is low, toy examples with large distances should be considered and sanity-checked by comparing both possible directions.

**Rec #3: Consider patching broken methods by switching spaces or metrics.** If a continual learning method fails in some particular application, it may be salvageable by altering the space in which distances are measured. Suppose, for example that one uses functional regularization in a task where the output of the network is target robot arm pose parameterized by joint angles. This may fail if task success is dependent on end effector pose, and the sensitivity of end effector pose to joint angle is itself highly dependent on robot pose, due to nonlinear kinematics. In this case, re-expressing the output in terms of end effector pose via a kinematics model may resolve these difficulties.

### 3.3. On Objectives

> **Considerations: Obj.** Current perspectives on continual learning tend to focus narrowly on accumulating knowledge through classification tasks. However, this view may be inherently limiting, as it emphasizes conditional knowledge ("which class, given these classes?") over unconditional understanding. The relationship between classification, density estimation, and generative modeling suggests broader ways to think about knowledge retention in continual learning systems.

**Cons #1: Accumulating unconditional knowledge.** The knowledge involved in successful classification is in-

herently of a very conditional nature, *i.e.*, we answer the question "given that this datapoint is drawn from the distribution of one of these $N$ classes, which class is it". We argue that focusing on classification objectives over density estimation or generative objectives makes continual or life-long learning unnecessarily overcomplicated. For example, out of distribution detection is clearly more closely related to density estimation, and there are whole classes of replay based continual learning algorithms which are closely related to generation. We believe that building continual learning algorithms on top of narrow classification tasks neglects the potential synergies of introducing generative or density based objectives, as we shall now discuss.

> **Recommendations: Obj.** Drawing from our analysis of the role of different learning objectives, we propose several directions for expanding beyond pure classification in continual learning. These recommendations emphasize the potential benefits of incorporating generative and density-based approaches, both for avoiding catastrophic forgetting and for more robust task identification.

**Rec #1: Consider generation for avoiding forgetting.**
Where the base task incorporates a generative objective, many challenges related to regularizing on or reviewing data examples from previous tasks are greatly simplified by direct exploitation of this generative function to create synthetic datapoints (Robins, 1995).

**Rec #2: Consider densities for task identification.**
In the presence of density estimation capabilities available from the base task, it is much easier to assign future datapoints to tasks and to consider questions of task boundaries, be they discrete or continuous.

**Rec #3: Consider the energy-based model connection.**
Even in the case of primarily classification objectives there seems to be great potential for density estimation via connections to energy-based models (Grathwohl et al., 2020; Li et al., 2022). This could be of great use in the primary evaluation settings common within continual learning.

## 4. Alternative Views

The most challenging response to our perspective holds that 1) it is convenient to study fundamental unsolved problems in CL in this setting, such as the stability gap (De Lange et al., 2023) or loss of plasticity (Dohare et al., 2024) and 2) that these problems are not specific to this setting and good solutions to classification benchmarks should generalize to other settings. We agree in principle, but are concerned that overreliance on this setting comes with the risk that the solutions so developed may be too specific (*e.g.*, through the use of task- or class-specific components), even if more

general solutions exist. Other alternatives hold further that incremental classification is rightly prioritized due to its simplicity and should therefore be solved first, however we believe this choice of focus has instead occurred mostly for historical reasons — it is not clear to us how classification is relevantly simpler or more principled than, *e.g.*, regression.

Secondly, while incremental classification is often used as a convenient test case for CL, it is widely accepted that a new CL algorithm should be demonstrated on problems more complex than, *e.g.*, Split-MNIST or Split-CIFAR. It is common, therefore, to expect CL works to include larger and more "real-world" benchmarks, such as ImageNet, and we expect the requested scale and associated compute requirements to escalate in the future. It is the position of this paper that there are more valuable and conceptually interesting sources of difficulty against which new algorithms may be tested, and we urge scholars to ask new algorithms to demonstrate generalizability beyond incremental classification, in lieu of simply scaling up classification benchmarks.

## 5. Concluding Remarks

We have argued that expanding the scope of continual learning (CL) research beyond supervised classification with discrete tasks is crucial for the development of theoretically grounded and widely applicable CL systems. Through the use of illustrative examples, we have analysed the limitations of naïvely applying current approaches, and have noted the potential of the notions of "task", "similarity" and "memorization" for generalization.

Key recommendations include selecting appropriate spaces in which to measure similarity, taking care when choosing distance measures on those spaces, and accounting for any relevant asymmetries. We further suggest integrating generative objectives for the mitigation of catastrophic forgetting and the potential of density modeling to identify task transitions and out-of-distribution data. By pursuing these research directions and examining the CL problem from the first principles when encountering atypical applications, we believe that the field can make significant strides towards flexible and adaptive learning systems that bring the recent progress of the field to new areas.

Although significant challenges remain in broadening CL beyond supervised classification, we believe the concrete recommendations in this paper — from careful selection of similarity metrics to integration of generative objectives — provide practical steps forward. By examining how current methods fail on non-standard problems and analyzing their underlying assumptions, we hope that this more nuanced view of CL's scope and challenges will help researchers develop methods that gracefully handle the diversity of tasks found in practice.

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
