# OpenReview forum: "Position: Continual Learning Should Move Beyond Incremental Classification"
_ICML.cc/2025/Position_Paper_Track — Submitted to ICML 2025 Position Paper Track_

### Official Review · Reviewer_b4sL · 2025-02-19

**Significance:** 4
**Argument Clarity:** 2
**Rating:** 1
**Confidence:** 4

**Questions:**

1. What is the purpose of the illustrative examples? For example, the facial expression classification example, how is it related to Continual Learning at all? Why do we need CL to solve this problem? Same question applies to the other two examples.

2. How do the considerations in Section 3 relate to your findings in Section 2? Can you make sure that every statement in Section 3 is supported by argument or evidence and that every expression you use has been properly introduced or defined? Some of them seem to be recommendations without justification, e.g. "Use drift processes": the only justification for this recommendation is "as this makes the temporal structure explicit", temporal structure of what? What is a "drift process"? What is a "continuous task space"?

3. What is a datapoint-indexed distribution?

4. Can you formalize the problem of continual learning? Can you map the illustrative examples to your formal description?

**Discussion Potential:**

3

**Paper Summary:**

The authors advocate for moving beyond the incremental classification paradigm in research on continual learning. They discuss three illustrative examples to highlight the problems of existing CL research. To support their position, they identify three fundamental challenges and provide recommendations to face them.

## update after rebuttal
I find the paper hard to read (long-winding sentences, mentioning things that have not been introduced, or ambiguous references), the figure quality is insufficient, and, as other reviewers have also mentioned, there are no case studies or other evidence to make a sufficiently novel or surprising point. At the current stage, I will therefore not recommend acceptance.

**Position:**

Yes

**Position In Title:**

Yes

**Related Work:**

3

**Strengths And Weaknesses:**

While the topic has promise to spark discussions in the community, the paper has some significant flaws in my view. A lack of formal problem definitions and in general a lack of precise formal statements is the main issue. This makes it hard to follow the illustrative examples, because the goal is not formalized and the problem setup remains vague. Consequently, it is hard to understand how the considerations and recommendations follow from the examples. The writing needs to be significantly improved in terms of clarity and general flow of reading. I generally find the direction of the paper promising.

Detailed Comments:
2.1
    the example is recognizing facial expressions. it is not clear to me how this is a problem of continual learning. Facial expressions are not something that evolve over time, angry today is the same as angry tomorrow, happy 100 years ago was the same as today.
    "It is not clear how to adapt CL methods that rely heavily on such clustering to this problem." I don't follow. The problem is a multi-class classification problem, it can be artificially adapted to CL just like any other multi-class classification problem. What I don't get is why this is necessary (Facial expression classification is not a continual learning problem)
    It sounds like you want to take a problem that is not a continual learning problem and make it into a continual learning problem. Furthermore this CL problem needs to be one that functions "correctly in the absence of class labels". I have no idea why anyone would want to do this and why class labels are important in doing this.
    "we would encounter similar issues" which issues are you referring to?
    "Alternatively, one could use the multi-target classes and consider a single cluster for the purposes of iCarl to correspond to a choice of class for every target" what are "multi-target classes", I don't understand that whole sentence

2.2
    As in 2.1, it is again not clear to me what the CL problem is. The prediction problem you describe is not a CL problem. Generating a trajectory for a robot arm is not a CL problem. Therefore, we do not need CL methods to solve this problem. But it seems that you are trying to make the problem into a CL problem. Why?

2.3 You describe classifying a digit as prime and divisible by 3 as different tasks whereas classifying different classes would be the same task, but different classes. This seems like an artificial distinction. "being prime" could also be viewed as a class and "divisible by 3" as another class, so then this problem would also be a class-incremental problem. I don't understand the point you are trying to make here.
You describe an example of pusing a box along the ground. As in the previous two subsection I must ask: How is this a continual learning problem? This problem is not going to change in 5 years, the box might change, the angle might change, but the problem will stay the same. It again sounds as if you are trying to create a problem where there is none.

2.4 the Zergling rush is described. Again, I don't see how this is a CL problem. Players need to memorize something, but that alone is not sufficient for CL.

Section 3:
You now present some considerations and recommendations. I recommend to try to connect this section to the previous one, Section 2. As it is I don't see how the two sections are connected. Also, the considerations are very vague and hard to follow. They should be supported by formal definitions and statements. E.g. Cons #2 Task Continuity: You are saying that a task-free setting should be explored, but it remains unclear why, what this means exactly, and how this can be useful.

Figures:
Figure 2 I see a red line on a ball, not sure what this should illustrate
Figure 3 it looks to me like the same image is shown 4 times in a row, I cannot see a box, the image is black with some white stuff which could be the robot arm

**Support:**

2

---

> ### Author Rebuttal · Authors · 2025-03-30
>
> We are grateful for the strong engagement with our work and strong score on significance, discussion potential and related work. While we cannot respond comprehensively within the character limit, we will gladly address any remaining points on request.
>
> Q1: Each example identifies a problematic task space, but does not necessarily suggest a specific sequence or stream of such tasks. Perhaps this is the source of confusion, and we will rectify it by adding such a suggestion to each relevant section. Examples below, with notes on other questions as necessary:
>
> 2.1: The mapping between felt emotion and facial expression is not constant across populations and cultures. One can easily imagine a CL problem where the task is recognizing emotions from facial expressions and the population changes, e.g. from contemporary China to archival footage of 1930s London.
>
> "Multi-target clusters": When adapting a 12-way multi-target binary classification problem for CL methods which require a partition of the dataset into a single set of clusters such as iCarl, then the only reasonable choice seems to be to take every possible combination of 12 binary choices to be a single cluster (which we call a "multi-target cluster"). Unfortunately we then have exponentially many clusters.
>
> We further consider a representation of faces in a two-dimensional continuous space, resulting in a linear regression problem for prediction. In this case there are no class labels, so a CL method would need to work correctly in their absence.
>
> 2.2: If I have a robot which successfully lays carbon fibres around the circumference of an aircraft body, and would like it to also be able to lay them lengthwise, then I have a CL problem on robot arm trajectories. The relevant manifold constraint here would be "remain exactly N millimeters from the surface of the aircraft body".
>
> 2.3: In order to push the box effectively, the robot must choose where to apply force. The correct location depends on the center of mass of the box. If the distribution of centers of mass changes between task A and B then we have a CL problem.
>
> The "artificial distinction" here is about mutual exclusivity of classes. {prime, not prime} and {divisible by three, not divisible by three} cannot be merged into a single target classification problem because data items would have multiple classes e.g. three is both prime and divisible by three. {4,5,6,7} and {8,9} however, can be so merged, as they are in the standard class incremental setting.
>
> 2.4: In competitive online starcraft the distribution of situations and strategies encountered changes over time, as the game is patched and different units become stronger or weaker, and as new strategies are discovered or go in and out of fashion. Consistently adapting to maintain high performance over time is therefore a CL problem.
>
>
>
> Q2a: We agree that section 3 could be more explicitly connected to section 2. We will be more explicit in highlighting the continuous task structure of 2.3 and 2.4 in section 3.1. We will refer back to the starcraft example when discussing temporal continuity in 3.1 and temporal correlations in the training data and schedule dependence in 3.2. We will refer back to the manifold constraints of 2.2 when discussing metrics in 3.2. In 3.3 we will refer back to 2.4 for generatively avoiding forgetting, and to 2.1 and 2.3 for using density estimation for task identification.
>
> Q2b: We will add an appendix describing drift processes and the related terms e.g. datapoint-indexed distribution in greater detail. The "temporal structure" in question here is the expected correlations in distribution of observed datapoints over time, we will make this more explicit.
>
> A "continuous task space" is a (not-necessarily finite) set of tasks for which the notion of "continuity" makes sense, i.e. a topological space whose points are tasks. For some of the discussion we may need this space to further be a smooth manifold. If the dimension of the task space is not finite, then we may need this to be a hilbert manifold. We are not confident that these distinctions would add clarity for the average reader, but will add an appendix describing which formal structures are required for which aspects of the discussion for the interested reader.
>
> Q3: A datapoint-indexed distribution associates to each datapoint a (potentially different) distribution from which it was sampled. A drift process is a way of associating to each datapoint a point in time, and to each point in time a distribution, such that we can require the distributions which are close in time to be similar. The latter makes the expected relationship between closeness in time and closeness in distribution explicit.
>
> Q4: We are wary of giving a formal definition of CL because it is difficult for the definition to be simple and yet also sufficiently expressive to cover all cases of interest. Indeed, overly narrow definitions likely contribute to the issue this paper criticizes.

---

### Official Review · Reviewer_E5mY · 2025-02-25

**Significance:** 3
**Argument Clarity:** 4
**Rating:** 4
**Confidence:** 4

**Questions:**

See Strengths And Weaknesses section.

**Discussion Potential:**

4

**Paper Summary:**

The paper makes the position that standard continual learning (CL) research should move beyond class incremental learning which is vastly prevalent in the literature. The authors recognize several axes in which it is challenging to adapt common methods or training paradigms. Specifically, accommodating to constraints, handling continuity in both time and the task space, and memorizing abstract notions instead of raw data. Then, the authors present three key challenges that should be addressed to move the field forward. (1) The nature of continuity in learning problems, (2) the choice of appropriate spaces and metrics for measuring similarity, and (3) the role of local objectives in learning. The authors describe the difficulties associated with each challenge and propose directions to solve or mitigate them.

**Position:**

Yes

**Position In Title:**

Yes

**Related Work:**

3

**Strengths And Weaknesses:**

Strengths:
- The paper deals with an important and highly relevant subject to the ML (and ICML) community.
- The authors back their claims with evidence from the literature and useful intuitions and use cases.
- The paper is written well and promotes the main message and research agenda in a coherent and clean manner.

Weaknesses/Questions:
- I believe one way to promote diversity of research in this field is by constructing relevant benchmarks and datasets. The authors touched on that point at the end of section 4, but I wonder if the authors can also reference to relevant studies that built such benchmarks.
- Adding on top of the challenges the authors stated I would like to add two more. (a) Continual learning in the presence of limited-sized datasets. Forming density estimation as the authors suggest in the presence of few examples can be challenging and I wonder how the authors suggest to overcome this difficulty. Perhaps the authors should also give special attention to that particular issue in the paper and reference relevant literature (e.g., [1]). (b) Continual learning from different modalities. Here there are several challenges related to learning well all the modalities, sharing information between modalities, varying modalities over time, etc. I think it will be valuable that the authors address that point as well (with relevant references).

In general I think the authors made a good job explaining and motivating their position. I do not have substantial comments and I recommend accepting this paper.

[1] Achituve, I., Navon, A., Yemini, Y., Chechik, G., & Fetaya, E. (2021, July). GP-Tree: A Gaussian process classifier for few-shot incremental learning. In International conference on machine learning (pp. 54-65). PMLR.

**Support:**

4

---

> ### Author Rebuttal · Authors · 2025-03-30
>
> Thank you for you engagement and feedback, we appreciate the positive remarks. Below we provide answers to your questions.
>
> Q1: We agree that constructing benchmarks and datasets is a good approach, and indeed argue for this. For many of our concerns we are unaware of such benchmarks already existing, but will be sure to include references to streaming datasets with non-discrete task boundaries, e.g. Zhipeng Cai et. al. 2021 Online continual learning with natural distribution shifts: An empirical study with visual data. (ICCV)
>
> Q2a: The reviewer is entirely right to point out the additional challenges associated to limited data. Presumably the difficulty with density estimation the author refers to would be overfitting to a point distribution on the data. In this case a more bayesian approach accounting for model complexity or for parameter uncertainty would be necessary. We note that in extreme cases of limited data it may no longer be true that storing a substantial fraction of the training data is impractical, as is often otherwise true in CL. The emphasis would instead shift to maximizing positive knowledge transfer between tasks. We will add a relevant discussion to section 3.
>
> Q3b: The opening of section 3.2 on spaces is a natural place to discuss this - in multimodal cases it is likely profitable to exploit the shared parameter spaces to transfer information instead of a potentially unshared function space. We will further consider how best to include multimodal challenges and relevant citations, perhaps in the section 2.4 commentary on memory. These difficulties are a natural fit for the paper, and we thank the reviewer for mentioning them - some such discussion was once included but later cut for space, fortunately it may now be re-included.

---

> > ### Comment · Reviewer_E5mY · 2025-04-04
> >
> > I am sorry for the late reply, I wrote a comment a couple of days ago but it wasn't visible to you. In any case, thank you for the answers. I believe the authors properly addressed my comments and those of the other reviewers; hence, I maintain my original score.

---

### Official Review · Reviewer_AzR8 · 2025-02-28

**Significance:** 2
**Argument Clarity:** 2
**Rating:** 1
**Confidence:** 4

**Questions:**

Could the authors provide statistical data or a sampling analysis to quantify the proportion of studies that focus exclusively on classification tasks? This would help substantiate the claim that CL research is overly concentrated in this area.

**Discussion Potential:**

2

**Paper Summary:**

This paper critically examines the dominant focus on incremental classification within the field of continual learning (CL). While acknowledging the role of incremental classification in advancing CL research, the authors argue that this narrow focus limits both theoretical development and practical applicability. Through concrete examples, the authors demonstrate the limitations of current CL approaches when applied beyond standard classification tasks. They identify three fundamental challenges: (1) understanding the nature of continuity in learning problems, (2) selecting appropriate spaces and metrics for measuring similarity, and (3) expanding learning objectives beyond classification. The paper provides specific recommendations to address these challenges, advocating for a broader scope of CL research.

## update after rebuttal

My major concerns (weaknesses 1 and 2) are not well addressed. In particular, given that continual learning studies are being actively extended beyond classification, the major claim of this study may not hold well.

**Position:**

Yes

**Position In Title:**

Yes

**Related Work:**

1

**Strengths And Weaknesses:**

**Strengths:**

1. The paper effectively highlights the risks of overemphasizing classification in CL research.
1. This paper offers a detailed examination of several key areas where current CL methods fall short, supported by concrete examples, which effectively underscores gaps in existing research.

**Weaknesses:**

1. While the paper presents compelling arguments, it lacks empirical validation or case studies to demonstrate the effectiveness of the proposed recommendations.
2. While the authors claim that CL research overly focuses on classification tasks, this perspective overlooks the significant body of work dedicated to other domains and tasks. For instance, there is substantial research addressing challenges in object detection [1] [2], instruction tuning for LLMs [3] [4], and reinforcement learning [5] [6]. These examples demonstrate that CL methods are being actively extended and adapted beyond classification.

[1] Liu, Yaoyao, et al. "Continual detection transformer for incremental object detection." CVPR 2023.

[2] Feng, Tao, et al. "Overcoming catastrophic forgetting in incremental object detection via elastic response distillation." CVPR 2022.

[3] Tan, Xiaoyu, et al. "Enhancing Task Performance in Continual Instruction Fine-tuning Through Format Uniformity." SIGIR 2024.

[4] Jiang, Gangwei, et al. "Unlocking the Power of Function Vectors for Characterizing and Mitigating Catastrophic Forgetting in Continual Instruction Tuning." ICLR 2025.

[5] Kaplanis, Christos, et al."Continual reinforcement learning with complex synapses." ICML 2018.

[6] Anand, Nishanth, et al. "Prediction and control in continual reinforcement learning." NeurIPS 2023.

**Support:**

2

---

> ### Author Rebuttal · Authors · 2025-03-30
>
> We thank the reviewer for reading our paper, and recognizing that our work "effectively underscores gaps in existing research" and "effectively highlights the risks of overemphasizing classification in CL research". We believe this constitutes a favorable evaluation of our paper in its role as a position paper. We would appreciate it if the reviewer treated our paper as such, reconsidering the weakness "presenting compelling arguments, it lacks empirical validation" (weakness 1).
>
> We now clarify the remaining questions and concerns below:
>
> Q1: Following Verwimp et. al. [7] we provide the requested analysis on their sampling of 77 continual learning papers (in which they studied consideration of memory constraints in the CL literature, based on papers explicitly tagged with "continual learning"). For the camera ready version we will extend this analysis to include  works published after [7], but did not have time to conduct additional rigorous sampling during the rebuttal window. We hope that reusing the already identified subset of CL papers from an existing published work makes our interim results here more persuasive.
>
> We conclude that 75% of these papers evaluate only on classification, and this increases to 86% if we include the closely related case of semantic segmentation (pixel-wise classification). Further, 88% consider only task/class/domain incremental settings (i.e. the space of tasks is discrete). If there are other specific questions the reviewer would like tested please let us know, and we will endeavor to do so in the remaining discussion time.
>
> W2: We argue that when designing *general* continual learning algorithms it is important to go beyond incremental classification. Indeed, the importance of this is underscored by the abstract of [1] Lie et. al.: "However KD and ER do not work well if applied directly to state-of-the-art transformer-based object detectors...". We thank the reviewer for this additional source of support for our central claims.
>
> We do not argue that it is inappropriate to design algorithms for a specific application *if generalization is not intended or desired*. All suggested papers except [5] are designing or adapting an existing CL algorithm for a particular application intentionally, and thus do not fall into the general CL methods literature we mean to discuss. In the case of [5], this is an interesting and well motivated parameter regularization method which unfortunately is only evaluated in RL settings despite likely having potential for generalization.
>
> We would like to emphasize that these many works designing special-case CL methods for particular applications are made necessary by the failure of the general CL literature to ensure broad applicability.  This is exactly why we decided to write this paper.
>
> W1: Regarding empirical validation of recommendations, we note that this is a position paper. It is responsive to the call for "...an argument in favor of or against a particular research direction...". It is not the intended function of a position paper to introduce and empricially evaluate particular methods.
>
> [7] Verwimp, Eli et. al. 2024 "Continual Learning: Applications and the Road Forward" (CoLLAs)

---

### Official Review · Reviewer_VRvY · 2025-03-13

**Significance:** 3
**Argument Clarity:** 3
**Rating:** 3
**Confidence:** 4

**Questions:**

1. Please clarify the meaning of "schedule dependence". I think CL algorithms should be expected to incorporate temporal correlations inherent in continuous streams of data.

**Discussion Potential:**

3

**Paper Summary:**

This paper advocates that continual learning research should move beyond the current focus on standard incremental classification tasks based on three concrete application examples. Three fundamental challenges are identified and for each challenge recommendations are made to advance the field.

**Position:**

Yes

**Position In Title:**

Yes

**Related Work:**

2

**Strengths And Weaknesses:**

Strengths:
+ This paper analyses the limitations of standard class-incremental continual learning through three concrete application examples. The use of illustrative examples makes the discussions intuitive to understand.
+ Three identified conceptual challenges: the nature of continuity, the choice of spaces and metrics, and the role of local objectives in learning, provide insights into the broader challenges of continual learning.
+ This paper is well-written and a pleasure to read.

Weaknesses:
- The scope of the continual learning literature used to establish the position is limited to the standard classification problems for i.i.d. data, overlooking recent advances on graph continual learning, which could provide additional perspectives on the discussed challenges.
- This paper lacks a critical discussion on fundamental challenges in current continual learning research. For example, it does not address how continual learning tasks should be configured to effectively validate algorithms or whether existing evaluation metrics are sufficient.

**Support:**

3

---

> ### Author Rebuttal · Authors · 2025-03-30
>
> Thank you for reading our paper and for the constructive feedback. We hope that the below answers will clarify your remaining concerns:
>
> Q1: We agree entirely with the reviewer regarding schedule dependence, and did not mean to imply otherwise. Section 3.1 Recommendation #2, to which we believe the reviewer refers, is intended only to say that CL algorithms should be invariant to that subset of schedule information which is uninformative. We will modify it to emphasize this further.
>
> W1: Thank you for the pointer to continual learning on graph data. Graph valued data were not our focus for this work, but we have found the recent survey [1] which shows some recent advances in the last 1.5 years. We will look through the relevant literature and consider how and where we can include relevant commentary in our work.
>
> W2: We claim that a major limitation of current evaluation metrics is the tendency to consider only one sort of CL problem, e.g. incremental image classification. We argue that a more diverse set of problems would avoid newly developed techniques being unnecessarily restricted in their applicability due to limiting design choices, e.g. assuming discrete tasks limits application. We make further recommendations about tasks, e.g. that methods should strive to be schedule independent to the maximum extent which makes sense in context.
>
> Perhaps we were insufficiently clear about the implication of these claims for evaluation. We will make sure to summarize the implications of our claims for which kinds of tasks and configurations should be included (e.g. a streaming setting with non-discrete tasks) to improve evaluation coverage. We consider other more general concerns about evaluation (e.g. replicability, hyperparameter searches, number of splits used in an incremental classification evaluation etc) to be beyond the scope of this paper.
>
> [1] Zhang Xikun et. al. 2024 "Continual Learning on Graphs: Challenges, Solutions, and Opportunities"

---

### Decision · Program_Chairs · 2025-04-27

**Decision:**

Reject

**Comment:**

In generally the direction of the paper looks promising. However while the authors claim that CL research overly focuses on classification tasks, this perspective overlooks the significant body of work dedicated to other domains and tasks, for instances, reinforcement learning, object detection, instruction tuning for LLMs etc. The writing needs to be significantly improved in terms of clarity and general flow of reading. Formal problem definitions with more precise formal statements can strengthen the claim and discussion better.